# Accurate Prediction of Sudden Cardiac Death Based on Heart Rate Variability Analysis Using Convolutional Neural Network

**DOI:** 10.3390/medicina59081394

**Published:** 2023-07-29

**Authors:** Febriyanti Panjaitan, Siti Nurmaini, Radiyati Umi Partan

**Affiliations:** 1Doctoral Program of Engineering Science, Faculty of Engineering, Universitas Sriwijaya, Palembang 30128, Indonesia; febriyanti_panjaitan@binadarma.ac.id; 2Faculty of Science and Technology, Universitas Bina Darma, Palembang 30264, Indonesia; 3Intelligent System Research Group, Universitas Sriwijaya, Palembang 30128, Indonesia; 4Faculty of Medicine, Universitas Sriwijaya, Palembang 30128, Indonesia; radiyati.u.p@fk.unsri.ac.id

**Keywords:** sudden cardiac death, heart rate variability, Convolutional Neural Network

## Abstract

Sudden cardiac death (SCD) is a significant global health issue that affects individuals with and without a history of heart disease. Early identification of SCD risk factors is crucial in reducing mortality rates. This study aims to utilize electrocardiogram (ECG) tools, specifically focusing on heart rate variability (HRV), to detect early SCD risk factors. In this study, we expand the comparison group dataset to include five groups: Normal Sinus Rhythm (NSR), coronary artery disease (CAD), Congestive Heart Failure (CHF), Ventricular Tachycardia (VT), and SCD. ECG signals were recorded for 30 min and segmented into 5 min intervals, following the recommended HRV feature analysis guidelines. We introduce an innovative approach to HRV signal analysis by utilizing Convolutional Neural Networks (CNN). The CNN model was optimized by tuning hyperparameters such as the number of layers, learning rate, and batch size, significantly impacting the prediction accuracy. The findings demonstrate that the HRV approach, in conjunction with linear features and the DL method, achieved a higher accuracy rate, averaging 99.30%, reaching 97% sensitivity, 99.60% specificity, and 97.87% precision. Future research should focus on further exploring and refining DL methods in the context of HRV analysis to improve SCD prediction.

## 1. Introduction

Sudden cardiac death (SCD) is a cardiovascular condition that is the primary cause of mortality among adults, affecting both individuals with a history of heart disease and those without [1]. SCD can strike people suddenly regardless of age and gender and is estimated to occur in 3 to 4.5 million people worldwide [2]. Therefore, the WHO considers SCD a critical issue in clinical cardiology that needs to be promptly addressed by every country [3].

Many factors contribute to the early identification of SCD. These factors include age, gender, race, genetic variables related to arrhythmia, coronary heart disease factors, and other structural factors of heart disease [4]. Electrocardiogram (ECG) tools can also be utilized to identify other factors associated with SCD [5]. ECG is a non-invasive device that measures and records the heart’s electrical activity, providing information on P waves, QRS complexes, and T waves [6]. It is a valuable tool in detecting heart abnormalities, diagnosing patient health, and assisting doctors in providing appropriate treatment [7]. The examination of ECG waveforms is utilized to identify different cardiac irregularities. The shape of the QRS complex is a critical aspect of analyzing ECG signals as it signifies ventricular depolarization [8]. Therefore, this study uses ECG analysis focusing on complex QRS waves and measuring R-R intervals indicative of heart rate variability (HRV). The predictive potential of the variability of consecutive R-R intervals appears highly promising [9,10]. As a result, researchers have expressed keen interest in detecting R-peaks from ECG signals and utilizing R-R variability or HRV for in-depth analysis [11,12]. The primary objective is to identify early SCD risk and mitigate global mortality rates associated with this condition.

Previous studies have explored using HRV to predict SCD across periods ranging from 1 to 60 min. While some studies achieved a maximum prediction accuracy of 91.67%, they were limited by their analysis, primarily due to the utilization of a restricted comparison group dataset. For instance, Ebrahimzadeh et al. (2019) [2], Devi et al. (2019) [3], Ebrahimzadeh et al. (2018) [13], and A. Parsi et al. (2020) [14] employed only a single comparison group in their investigation. In contrast, a recent study by Rohila et al. (2020) [15] incorporated a broader range of comparison group data, including normal subjects (NSR) and individuals with cardiovascular diseases such as congestive heart failure (CHF) and coronary artery disease (CAD).

In Ref. [15], the authors focused on identifying SCD using HRV and involved four comparison groups. In our study, we extended this research by expanding the number of comparison groups to five, which included individuals diagnosed with Ventricular Tachycardia (VT). VT is a cardiovascular condition with a high probability of SCD [13,16]. VT has been used in predicting SCD risk with 90.2% accuracy [14]. Therefore, this study has five comparison groups: NSR, CAD, CHF, VT, and SCD.

This study presents an innovative approach to analyzing HRV signals using deep learning (DL) methods. DL methods were employed to analyze HRV features to improve the early detection of lethal cardiac arrhythmia. Previous studies have utilized various Machine Learning (ML) algorithms such as multilayer perceptron (MLP), k-Nearest Neighbor (KNN), Support Vector Machine (SVM), Decision Tree (DT), and Random Forest (RF) to predict the risk of SCD. However, DL techniques, particularly Convolutional Neural Networks (CNN), have gained popularity in medical diagnosis due to their superior performance and effectiveness compared to traditional ML algorithms [17]. CNN have exhibited promising outcomes in conducting ECG classification among numerous deep-learning algorithms. The CNN model showcased remarkable overall performance, from 93.53% to 99% [18,19,20]. CNN is renowned for its proficiency in pattern recognition, combining feature extraction, dimensionality reduction, and classification techniques. The CNN method consists of three main layers: input, hidden, and output, with deeper layers enhancing the detection and classification of heart disease data [21]. In this study, the CNN method was optimized by tuning hyperparameters, including the number of layers, learning rate, and batch size, as these parameters significantly influence the model’s prediction accuracy.

This research makes the following contributions:The development of a 1D-CNN model to predict the risk of SCD based on HRV Features.The expansion of the number of comparison subjects in predicting SCD, that is, CAD, VT, CHF, and NSR.

## 2. Materials and Methods

This research employs a methodology utilized by multiple previous researchers in predicting risk SCD, i.e., using ECG patterns. A previous study utilizing a similar method was conducted [2,3,15]. The research steps undertaken in this study are outlined as follows:

### 2.1. Database

This study included five subject groups, namely NSR, CAD, CHF, VT, and SCD, with 115 subjects comprising 18 NSR, 51 CAD, 15 CHF, 11 VT, and 20 SCD cases. The ECG signals for these groups were obtained from the PhysioNet database [22]. We have chosen CHF, CAD, and VT as comparison groups due to their relevance to cardiac disease and their association with SCD. Including these groups provides a valuable context for evaluating the predictive capabilities of our model. In addition, the dataset on CHF and CAD has also been utilized by Rohila et al. in their study on SCD prediction.

For the SCD group, the duration of the ECG signal used was 30 min preceding the onset of VF. The primary objectives of this study were to identify individuals at risk of SCD and differentiate subject groups with non-SCD conditions (NSR, CAD, CHF, and VT). Non-SCD subjects were confirmed to be free from VF. While each group may have had more than two channels, this study focused on analyzing the first channel as it provided more reliable output signals [23]. The comparison data included signal durations from all five groups, each spanning 30 min. Specifically, the ECG signal of the SCD group was obtained 30 min before the onset of VF. The choice of a 30 min ECG duration was motivated by the maximum period of VT, which is 30 min or 1800 s [24].The 30 min ECG duration was segmented into 5 min intervals to facilitate analysis, resulting in six segments per subject. Using 5 min segments aligns with the recommended practice for HRV feature analysis [11].

### 2.2. Preprocessing

The 30 min ECG signal duration was divided into segments with a period of 5 min, resulting in six segments for each subject. A series of steps were followed to extract the R-R interval from the 5 min ECG signal. Firstly, noise and baseline wandering were eliminated using the Discrete Wavelet Transform (DWT) method, which involves passing the signal through low- and high-pass filters. This study employed the sym5 wavelet family for DWT [25]. Secondly, the ECG was normalized using two techniques: Normalized Absolute Deviation (NADev) and Normalized Absolute Difference (NADiff) [26]. Lastly, R-peaks were detected using the Hamilton–Tompkins algorithm [27]. This preprocessing methodology has been widely adopted by other researchers [28,29]. Figure 1 shows a comparison of ECG Signals before and after preprocessing in first segments for the five subject groups.

Figure 2 presents enlarged sample images with one minute of segmentation for SCD subjects. In panel (a) of the figure, ECG segment images are shown before preprocessing, showing the raw input data with noise. Panel (b) shows the ECG segment images after preprocessing, specifically with the identification and marking of the R peak.

### 2.3. Feature Extraction

After identifying the R-peaks, which correspond to the highest points in the ECG waveform representing ventricular depolarization, the next step is calculating the time intervals between successive R-peaks to obtain the RR intervals. Subsequently, various statistical measures are applied to the RR interval series to extract meaningful features. In HRV analysis, the following commonly computed features are considered:

The mean value of the RR interval (MeanRR).
(1)$$MRR=\frac{1}{N}\sum RR_{i}$$

The square root of the mean of the squared differences between adjacent RR intervals (RMSDD).
(2)$$RMSSD=\sqrt{\frac{1}{N}}\sum_{i=1}^{N-1}{\left({RR}_{i+1}-{RR}_{i}\right)}^{2}$$

The percentage of adjacent RR intervals differs by more than 50 ms (PNN50).
(3)$$PNN50=\frac{num(RR>50)}{num(RR)}$$

The standard deviation of all RR intervals (SDRR).
(4)$$SDRR=\sqrt{\frac{1}{N}\sum {\left(RR_{i}-{RR}_{m}\right)}^{2}}$$

The coefficient of variation of RR intervals (CVRR).
(5)$$CVRR=\frac{SDRR}{MeanRR}$$

The number of consecutive RR interval pairs that differ by more than 50 ms (NN50).
(6)$$\mathrm{Num}(\mathrm{abs}({RR}_{n+1}-{RR}_{n}))\geq 50\;ms$$
where N = Number of RR intervals; RRi = RR interval after; and RR_i + 1_ = RR interval before.

In addition to the six features mentioned above, this study includes two other features, namely the minimum RR interval and the maximum RR interval.

The results of this feature extraction process are documented in detail in Table 1. The table provides a comprehensive overview of the extracted HRV features, allowing for easy comparison and analysis across the different groups. The HRV features captured essential information relating to heart rate variability and served as valuable indicators for assessing autonomic nervous system activity and cardiac health.

### 2.4. Classification

This study employs a 1D-CNN architecture for analysis. CNN, or Convolutional Neural Network, is a deep neural network model with multiple stacked layers designed to mimic the activity of neurons in the human brain [30]. The CNN extraction process involves several hidden layers, including convolutional layers, ReLU activation functions, and pooling layers [31]. The CNN classification process includes a fully connected layer and an activation function (sigmoid or softmax) that produces the classification output [32].

Considering the part of the CNN method that can classify features extraction from HRV, this study proposes a Wavenet model. This model was proposed by Yue Meng et al. (2022) [33] and has successfully classified diseases with good performance accuracy. Wavenet is a generative model consisting of residual blocks with gated activation. With regard to the equation form of the Wavenet model [34], it can be seen in the following equation.
(7)$$z=\tanh\left(Wf,k*x\right)⊙\sigma (Wg,k*x)$$
where z represents the unit of gated activation, * is the operator of convolution, ⊙ multiplication function operator, σ(.) the sigmoid function, k is the number of layers, f is the filter, g is the gate, W is the learnable convolution filter, and x is the waveform [34,35].

Our proposed model’s HRV feature input sample depends on the previous output sample. The feedback from each predicted sample helps the network predict the following sample. Dilated convolutional layers are employed in the network to enable large skips of input data, increasing the receptive field for better results. Only a few layers enhance the receptive fields to maintain input resolution and ensure output data consistency on a larger time scale [34]. The distribution of each HRV feature sample is achieved using softmax. The residual and parameterized skip connections accelerate convergence and facilitate intense model training. Our proposed Wavenet model is illustrated in Figure 3.

Description of our proposed method is as follows:Input layer: The input to the network is a sequence of the length 9.Reshape layer: reshapes the input sequence to have a shape of (9, 1), converting it into a 1D Signal.Conv1D layers: There are several hidden layers in Conv1D layers, each with different dilation rates. The dilation rate determines the spacing between the values in the kernel. Higher dilation rates allow the network to capture larger patterns while retaining fewer parameters. The Conv1D layers have 64 filters each and a kernel size of 2. They use the “causal” padding, which ensures that the output at each time step only depends on the current and past inputs. The dilation rates for these layers are based on the values in the dilatation_rates list, which are powers of 2 ranging from 1 to 64. The activation function used in these Conv1D layers is ReLU.Multiplication and activation layers: Following each Conv1D layer, two activation functions are applied independently: tanh and sigmoid. The outputs of these activation functions are then element-wise multiplied together.TimeDistributed Dense layer: Each output from the multiplication and activation layers is passed through a TimeDistributed Dense layer with 64 units and ReLU activation. The TimeDistributed layer applies the same Dense layer to each time step of the sequence.Skip Connections: The outputs from the TimeDistributed Dense layers are appended to a list called skips. These skip connections allow information from earlier layers to be propagated and combined with later layers.Add and Activation layers: The elements in the skips list are summed using an Add layer. Then, the ReLU activation function is applied to the summed output. After the Conv1D layers, the code continues with other layers, such as Flatten, Dropout, Dense, and the final output layer. This model uses the 1D-CNN Wavenet architecture with a total of 11 layers (Table 2).

This model has been optimized because this process is crucial to ensure that the test results do not suffer from overfitting. The optimized hyperparameters are the hidden layers, learning rate, and batch size (Table 3). The grid search process was conducted comprehensively through predefined subclasses of model hyperparameter combinations [36].

The hyperparameters were trained and evaluated using the first 5 min segments. The model with the lowest loss and highest accuracy was selected as the best-performing model. Once the best model was determined, it was tested on five other segments from each subject group. Each segment was compared to obtain performance predictions for each class group. The first 5 min before the occurrence of SCD were compared with the first 5 min of the other classes, and this process was repeated for the remaining segments. The average accuracy performance was calculated using a confusion matrix. There were 115 subjects, with a training distribution of 80% and a testing distribution of 20%.

The performance of the classification model is evaluated using a confusion matrix [37]. Each matrix column displays the class prediction results, while each row shows the classified class results. Accuracy, sensitivity, specificity, and precision are used to evaluate the performance of the proposed model.

The experiment runs on a computer platform with specifications including intel Core i7-8750H Ram 16Gb NVIDIA GTX 1080 with 4GB GDDR5 graphics memory and operating system Windows 10. DL structure developed using Python Programming with computation utilizing TensorFlow-GPU with Keras Neural Network Library.

## 3. Results

This study involved six segments extracted in HRV for each subject. The technique for extracting HRV features from the ECG signal is described in the pre-processing stage and feature extraction. This research hyperparameter tuning was conducted using the grid search method for each hidden layer (3, 5, 7, 9), utilizing a list of learning rate and batch size values (Table 3). Based on these four hidden layers, the study resulted in four models. Figure 4; Figure 5 show the loss and accuracy for four models using 140 epochs.

Figure 4 presents the loss values for four models, showcasing their performance during the training and testing phases. The loss serves as a measure of dissimilarity between the predicted and actual outputs of the models. Regarding training loss, Model 1 initiates with a value of 0.0346 and consistently decreases over the epochs. Model 2 starts with a higher training loss of 0.0693 but follows a similar descending trend. Model 3 exhibits a higher initial training loss of 0.513 but undergoes rapid reduction as the training progresses. Model 4 begins with a low training loss of 0.0248 and steadily declines. Shifting attention to the testing loss, Model 1 begins at 0.041 and maintains a relatively stable pattern throughout the evaluation. Model 2 starts with a higher testing loss of 0.0625 but follows a similar trend to Model 1. Model 3 commences with a testing loss of 0.052, remaining consistently low. Lastly, Model 4 demonstrates a testing loss of 0.060, which remains stable across the epochs. The graph visually captures the trajectory of loss values during training and testing, providing valuable insights into the models’ learning progress and their ability to generalize to the test data.

Figure 5 showcases the accuracy values for four different models, illustrating their performance in terms of training and testing. Accuracy represents the percentage of correct predictions made by the models. Regarding training accuracy, both Model 1 and Model 2 start at a high accuracy of 98.91%. These models maintain consistent accuracy throughout training, suggesting stable and effective learning from the training data. Model 3 and Model 4, on the other hand, start with perfect accuracies of 100%. These models exhibit no misclassifications during training, indicating their ability to learn the training data perfectly. Moving on to the testing accuracy, Model 1 begins with an accuracy of 99.42%, maintaining a high level throughout the evaluation. Model 2 starts slightly lower with an accuracy of 99.13% but demonstrates strong performance on the test data. Model 3 maintains a perfect accuracy of 100% during testing, indicating its ability to generalize well beyond the training data. Similarly to Model 3, Model 4 also achieves a perfect accuracy of 100%, showcasing excellent performance on the test data.

Model 3 is the best model due to its perfect accuracy, low training loss, and low testing loss. Its ability to generalize and consistently make accurate predictions on the training and testing datasets makes it a reliable choice for the given task. The best-performing model had seven hidden layers, a learning rate of 0.001, and a batch size of 128. This model was trained and tested using the first segment at a five-minute interval. This resulted in four models, as shown in Table 4.

Table 4 presents the performance of a grid search to determine the optimal hyperparameters for CNN, including the range of hidden layers, learning rate and batch size. The number of hidden layers plays a crucial role in the network’s capacity to learn complex patterns and relationships in the data. However, too many layers may lead to overfitting, while too few may result in underfitting. Each combination is trained and evaluated using a predefined metric, such as accuracy or loss. The model that achieves the highest performance on the validation set is then selected as the best-performing model.

The best hyperparameters mentioned above are optimal based on evaluating multiple models with different hyperparameter combinations. Through this iterative process, the grid search identifies the hyperparameters that yield the best performance on the testing set.

The 1D-CNN Wavenet architecture of the best model (model 3), including the number of layers, layer parameters, and output shape, is presented in Table 5. This table provides a comprehensive overview of the model’s structural components, allowing for a detailed understanding of its configuration and architecture.

Figure 6 shows the performance of a selected model in training and testing. The left-hand side of the figure shows the training and testing loss for 140 epochs. The right-hand side of the figure shows the training and testing accuracy for 140 epochs. The graph demonstrates that training and testing loss exhibit significant reduction and stabilization after several epochs, ultimately reaching a value of approximately 0.0052. The training and testing accuracy of the model comes to 100% after 140 epochs, implying that the model can precisely classify the input data into their respective categories. This high accuracy suggests that the model performs effectively and can be applied for predictive purposes.

Following the careful selection of Model 3, our study proceeds by leveraging this chosen model to analyze the 2nd and 6th segments. Table 6 and Figure 7 present the performance evaluation of a classification model that effectively predicts SCD using six segments for all subjects. In the first segment, the model incorrectly classified SCD as VT. Similarly, in the third segment, the model misclassified VT as SCD. The prediction errors in these cases may have occurred due to the similarity in morphology between these types of rhythms. Overall, the model performed well, with only two incorrect predictions. However, it accurately predicted all instances in the other four segments. The performance evaluation was based on commonly used metrics such as accuracy, sensitivity, specificity, and precision, which are important for assessing the performance of classification models.

The confusion matrix (Figure 7) represents the performance of the classification model in predicting SCD for each of the six segments. The confusion matrix also reflects the two wrong predictions mentioned in the table (Table 6).

From Figure 7, the model’s average accuracy in predicting SCD risk was 99.30%. This indicates that the model achieved high overall correctness in its predictions. However, it is worth noting that more than accuracy is needed to provide a complete picture of the model’s performance, and other metrics should also be considered. The model’s sensitivity was calculated to be 97%, representing the proportion of correctly identified individuals at risk of SCD out of all the actual cases. This shows that the model had a high ability to detect individuals at risk of SCD. The specificity of the model was found to be 99.60%. This metric measures the proportion of correctly identified individuals without the risk of SCD out of all the non-SCD cases. The high specificity indicates that the model performed well in correctly classifying individuals who were not at risk of SCD. The precision of the model was determined to be 97.87%. Precision represents the proportion of correctly identified individuals at risk of SCD out of all the predicted positive cases. This indicates that the model had a high level of accuracy in correctly identifying individuals who were truly at risk of SCD among the predicted positive cases.

Overall, these results demonstrate that the optimized CNN model, utilizing HRV analysis, achieved a high accuracy and performance in the early prediction of SCD. The model showed strong sensitivity, specificity, and precision, indicating its potential usefulness in identifying individuals at risk of SCD. Performance for each class subject is shown in Table 7.

## 4. Discussion

Furthermore, we introduced eight linear features for HRV analysis, which were used to analyze quality ECG signals. This research is the first to use CNN to extract HRV features for predicting SCD risk in five comparison groups with six segments.

Our study has effectively employed Model 3 to analyze multiple segments and predict SCD. However, we have observed misclassifications in certain segments, likely due to the similarity in morphology between SCD and VT rhythms. These findings highlight the challenges associated with classifying ECG signals based solely on morphology and emphasize the need for further research to explore alternative approaches that can enhance rhythm classification accuracy with additional HRV features.

This study offers a novel approach to detecting the risk of SCD. The studies analyzing the risk of SCD have been summarized (Table 8). Our study emphasizes the analysis of HRV signals’ short duration to predict SCD; the HRV duration of 30 min before VF onset was analyzed by dividing it into short-term segments. The use of HRV segments with a period of 5 min from the ECG signal is necessary to detect and reduce bias due to the selection of data with a length of 30 min duration. Overall, our findings highlight the potential of CNN in extracting HRV features to predict SCD risk in a diverse population accurately.

Our study represents a novel approach by utilizing CNN to extract HRV features for predicting SCD risk. While previous studies have investigated various methods of ML for SCD risk prediction, the application of CNN specifically for extracting HRV features in this context has been explored. By leveraging the power of CNN, we aim to capture complex patterns and relationships within the HRV data, potentially leading to improved predictive performance. To highlight the potential superiority of our method, a thorough comparison with previous studies is essential. We extensively review the existing literature on SCD risk prediction and discuss the methodologies, features, and performance metrics used in those studies (Table 8); by comparing our approach to these established methods, we can demonstrate the advantages and strengths of utilizing CNN for HRV feature extraction by optimizing the model with hyperparameter tuning.

The proposed model uses a 1D-CNN Wavenet model. In optimizing the model to separate individuals at risk of SCD, this study has conducted hyperparameters tuning, and the selected parameters use seven hidden layers, a 0.001 learning rate, and a batch size of 128. This model strengthens our findings in classifying SCD risk with an average accuracy of 99.30%, achieving 97% sensitivity, 99.60% specificity, and 97.87% precision.

Additionally, one aspect that sets our research apart is utilizing a larger dataset. By incorporating a substantial amount of data, we can increase the robustness and generalizability of our findings. This larger dataset provides an opportunity to train the CNN model, potentially enhancing its ability to learn and extract meaningful HRV features for accurate SCD risk prediction.

The performance of the proposed method analyzed with the test data appears promising for identifying SCD risk. However, some limitations need to be discussed. First, our study used limited patient information and possibly a very heterogeneous group, but the SCD subject data provide important pathogenic details on sudden death. Second, although the results for SCD risk classification seem promising, our study only used linear features. In contrast, HRV features provide two other features: time-frequency and non-linear features. In the future, the challenge is to use more available HRV features and add a comparison group to predict SCD risk.

## 5. Conclusions

This study employed a 1D-CNN Wavenet model and utilized a larger comparison group dataset than that of previous studies on SCD prediction. Through hyperparameter tuning, the model achieved high accuracy, sensitivity, specificity, and precision in classifying SCD risk. While the performance of the proposed method in identifying SCD risk appears promising based on the test data, there are limitations to consider. The study relied on linear HRV features, and future research should explore including time-frequency and non-linear features to improve prediction accuracy. Additionally, the study emphasized the need for more comprehensive patient information and the incorporation of a comparison group to enhance SCD risk prediction. Overall, the findings demonstrate the potential of using CNN-based models and HRV analysis to accurately predict SCD risk, offering a novel approach for identifying individuals at risk of SCD in diverse populations.

The proposed model’s CNN has the capability to perform real-time classification of ECG signals, indicating its potential for implementation in clinical settings. Moreover, our algorithm offers the advantage of being cost effective.

## Figures and Tables

**Figure 1 medicina-59-01394-f001:**
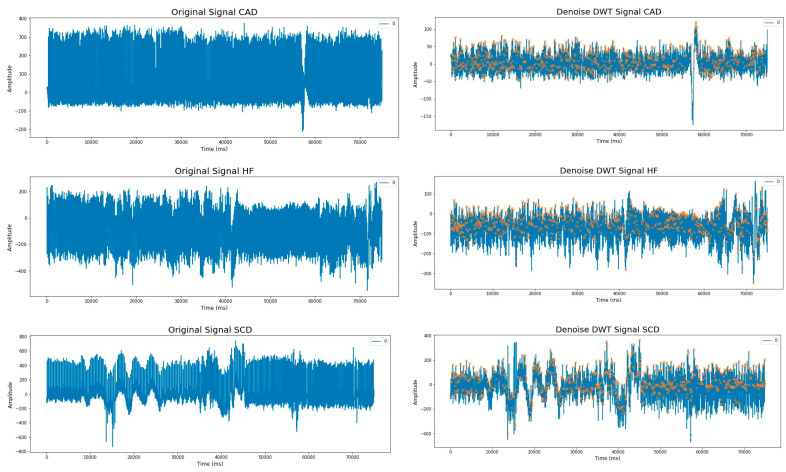
(**a**) ECG Signal with noise; (**b**) ECG Signal after preprocessing with R-Peaks are highlighted with a yellow "x" label.

**Figure 2 medicina-59-01394-f002:**
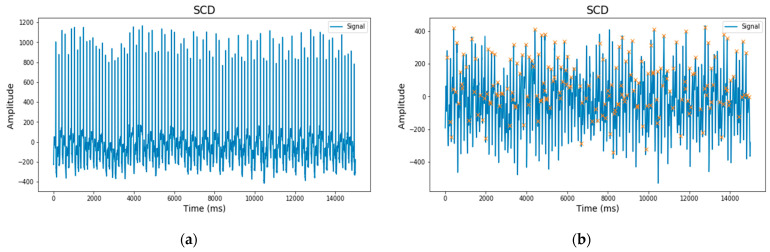
Sample picture for one minute interval for subject SCD; (**a**) ECG signal with noise; (**b**) ECG signal after preprocessing with R-peaks are highlighted with a yellow "x" label.

**Figure 3 medicina-59-01394-f003:**
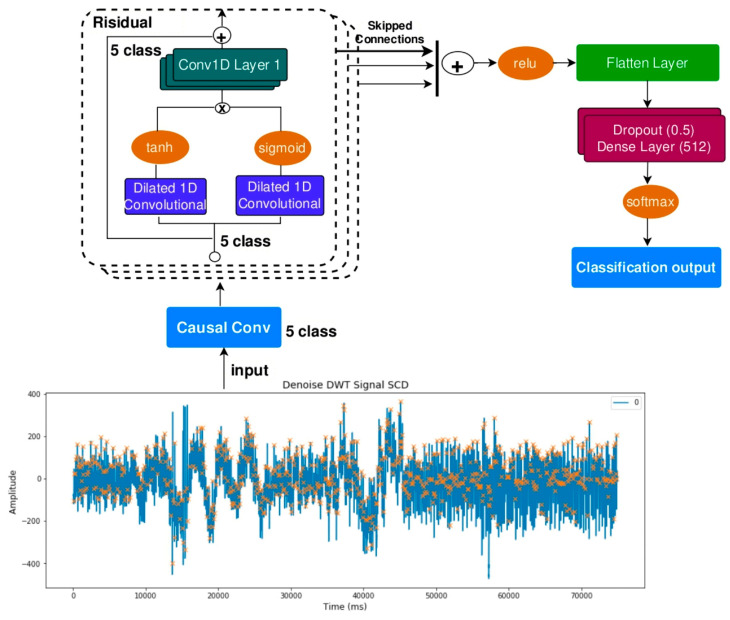
1D-CNN in proposed method used Wavenet.

**Figure 4 medicina-59-01394-f004:**
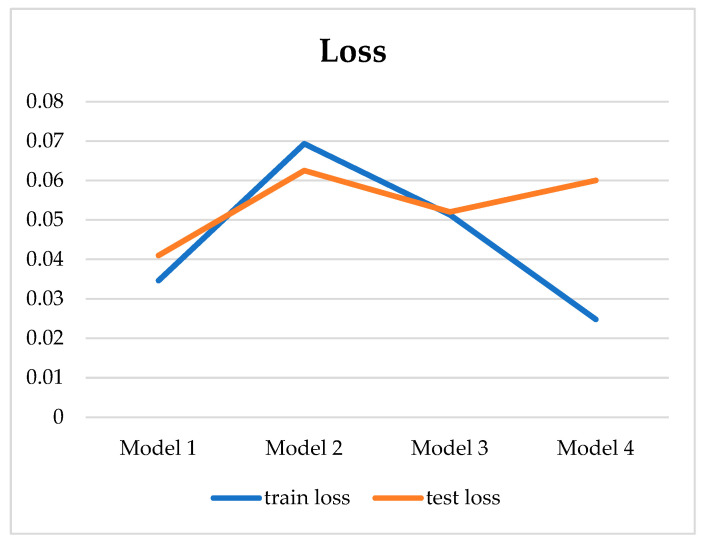
Loss train dan test for four models.

**Figure 5 medicina-59-01394-f005:**
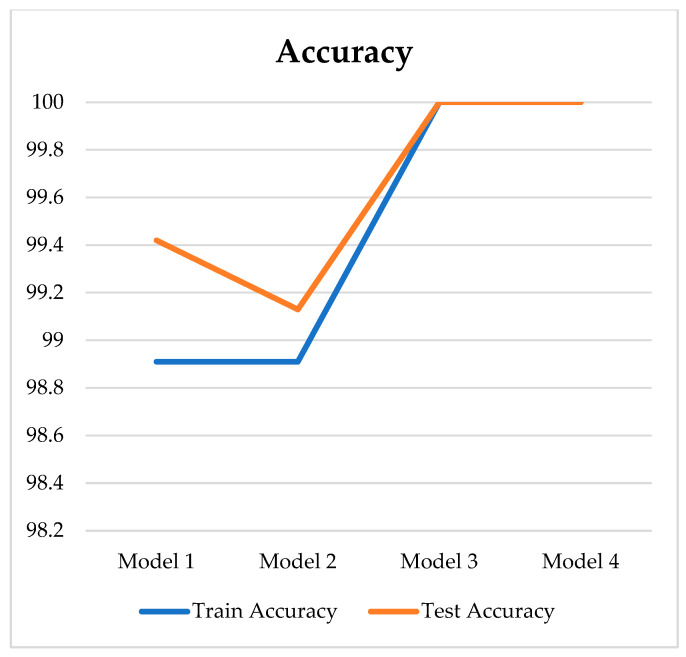
Accuracy train dan test for four models.

**Figure 6 medicina-59-01394-f006:**
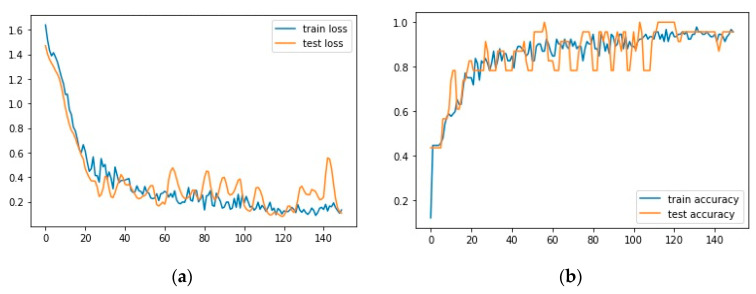
Graph of training dan testing the selected model; (**a**) loss; (**b**) accuracy.

**Figure 7 medicina-59-01394-f007:**
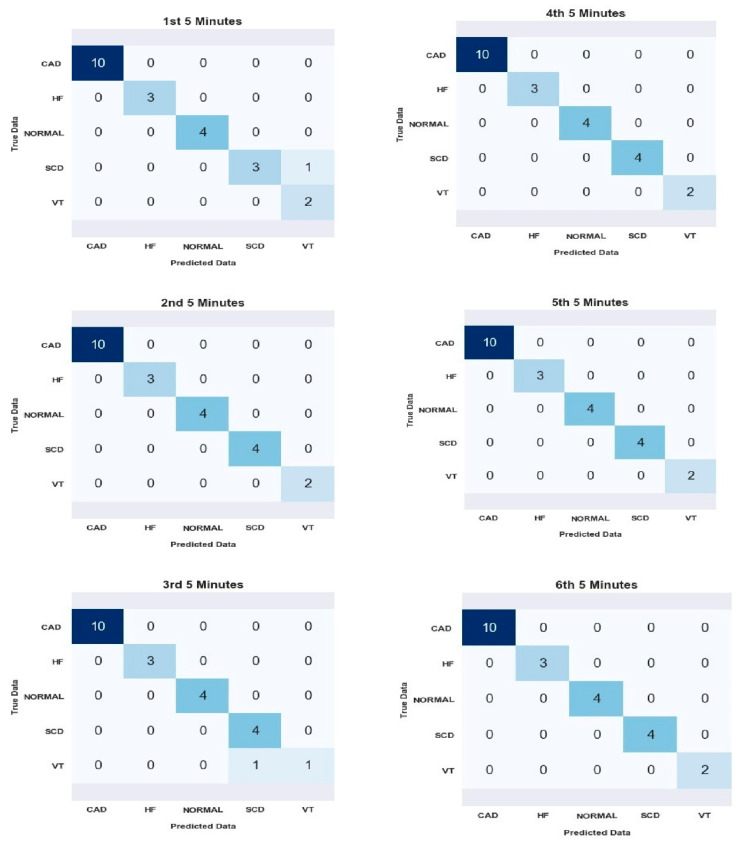
Confusion matrix results for all segments analyzed.

**Table 1 medicina-59-01394-t001:** HRV features five subjects.

Class	Patient	Time-Domain Features							
		MRR	SDRR	RMSSD	PNN50	CVRR	NN50	Min_RR	Max_RR
CAD	20031	0.977	0.722	0.897	0.009	0.739	0.951	0.007	4.687
HF	Chf01	0.787	0.425	0.581	0.009	0.540	0.928	0.007	2.617
NSR	nsr001	0.769	0.315	0.442	0.005	0.409	0.586	0.412	2.078
SCD	30	0.736	0.482	0.632	0.009	0.655	0.919	0.007	3.578
VT	106	0.906	0.684	1.035	0.009	0.754	0.955	0.007	3.960

**Table 2 medicina-59-01394-t002:** The number of layers from our proposed method.

No.	Layer	Number of Layers
1	Input	1
2	Reshape	1
3	TimeDistribute Dense (Loop)	hidden layers
4	Conv1D (Loop)	hidden layers
5	Multiply	1
6	TimeDistribute Dense	1
7	Add (Loop)	hidden layers
8	Flatten	1
9	Dropout	2
10	Dense	2
11	Output	1

**Table 3 medicina-59-01394-t003:** Optimized hyperparameters.

Hyperparameters	List of Values
Number of hidden layers	3, 5, 7, 9
Learning rate	0.1, 0.01, 0.001, 0.0001
Batch Size	32, 64, 128, 256

**Table 4 medicina-59-01394-t004:** Performance of CNN Wavenet with best hyperparameter for 5 min on test data.

Model	Number of Hidden Layers	Learning Rate	Batch Size	Epoch	Loss	Acc. (%)	Sens. (%)	Spec. (%)	Pres. (%)
1	3	0.001	32	140	0.041	99.42	96.67	100	98.46
2	5	0.0001	32	140	0.060	99.13	95.83	99.49	96.58
3	7	0.001	128	140	0.052	100	100	100	100
4	9	0.0001	32	140	0.060	100	100	100	100

**Table 5 medicina-59-01394-t005:** 1D-CNN Wavenet architecture with best hyperparameter.

No.	Layer	Layer Parameter	Outpshape
1	Input	shape = (9,)	(None, 9)
2	Reshape	target_shape = (9, 1)	(None, 9, 1)
3	TimeDistributed (Dense)	units = 64, activation = ‘relu’	(None, 9, 64)
4	Conv1D	filters = 64, kernel_size = 2	(None, 9, 64)
5	Conv1D	filters = 64, kernel_size = 2	(None, 9, 64)
6	Activation	activation = ‘tanh’	(None, 9, 64)
7	Activation	activation = ‘sigmoid’	(None, 9, 64)
8	TimeDistributed (Dense)	units = 64, activation = ‘relu’	(None, 9, 64)
9	… (repeated dilations)	-	-
10	Add	-	(None, 9, 64)
11	Activation	activation = ‘relu’	(None, 9, 64)
12	Flatten	-	(None, 576)
13	Dropout	rate = 0.5	(None, 576)
14	Dense	units = 512, activation = ‘relu’	(None, 512)
15	Dropout	rate = 0.5	(None, 512)
16	Dense	units = 512, activation = ‘relu’	(None, 512)
17	Dense	units = 512, activation = ‘relu’	(None, 5)

Note: The None dimension represents the batch size.

**Table 6 medicina-59-01394-t006:** Performance of classification using 1D-CNN Wavenet model (each segment).

Segments	Average for All Subject			
	Acc. (%)	Sens. (%)	Spec. (%)	Pres. (%)
1st	98.26	95	99.05	93.33
2nd	100	100	100	100
3rd	98.26	90.00	98.95	96.00
4th	100	100	100	100
5th	100	100	100	100
6th	100	100	100	100

**Table 7 medicina-59-01394-t007:** Performance of classification using 1D-CNN Wavenet model (each class).

Subject	Acc. (%)	Sens. (%)	Spec. (%)	Pres. (%)
CAD	100	100	100	100
HF	100	100	100	100
SNR	100	100	100	100
SCD	99.28	95	99.05	99.33
VT	99.28	90	98.95	96.00
average	99.30	97	99.60	97.87

**Table 8 medicina-59-01394-t008:** Comparison of results with state-of-the-art studies.

Authors (Year)	Signal Length Prediction Period	No. of Subject	Features	Method	Performance Result (%)			
					Acc.	Sens.	Spec.	Pres.
Ebrahimzadeh et al. (2018) [13]	12 min before 1 min interval	35 NSR 35 SCD	HRV Time-Domain, Frequency-Domain, Time-Frequency, Non-Linear	MLP	83.87	82.66	85.09	84.72
Ebrahimzadeh et al. (2019) [2]	13 min before 1 min interval	30 NSR 35 SCD	HRV Time-Domain, Frequency-Domain, Time-Frequency, Non-Linear	MLP	84.28	85.71	82.85	83.33
Devi et al. (2019) [3]	10 min before 5 min interval	18 NSR 15 HF 18 SCD	HRV Classical, non-linear, and CWT features	KNN	83.33	75	87.5	75
Ashish Rohila et al. (2020) [15]	1 hour before 5 min interval	18 NSR 23 CAD 15 HF 20 SCD	HRV Entropy, Poincare plot, S-transform	SVM and DT	91.67	83.33	94.64	84.75
A. Parsi et al. (2020) [14]	5 min before 5 min interval	106 VT 29 VF	HRV Time-Domain	SVM KNN RF	90.2	88.8	94.2	-
Our Work	30 min before 5 min interval	18 NSR 51 CAD 15 HF 11 VT 20 SCD	HRV Time-Domain	1D-CNN Wavenet model	99.30	97.00	99.60	97.87

## Data Availability

Dataset utilized in this study is freely available and can be downloaded from the following websites: https://physionet.org/content/sddb/1.0.0/ (SCD); https://physionet.org/content/mitdb/1.0.0/ (VT); https://physionet.org/content/chfdb/1.0.0/ (CHF); https://physionet.org/content/ltstdb/1.0.0/ (CAD); and https://www.physionet.org/content/nsrdb/1.0.0/ (NSR) (accessed on 5 February 2021).

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
