# Peer review of "Accurate Prediction of Sudden Cardiac Death Based on Heart Rate Variability Analysis Using Convolutional Neural Network"

_medicina, 2023, doi:10.3390/medicina59081394_

Round 1

Reviewer 1 Report

-It is not necessary to explain an abbreviation each time you use. For instance, you explain Heart Rate Variability 60 (HRV) for several times.

-In Material and methods authors acclaimed that “a standard approach”. Please define this approach and cite a reference for it.

Author Response

Dear Reviewer, Thank you for granting us the opportunity to submit a revised draft of our manuscript titled "Accurate Prediction of Sudden Cardiac Death Based on Heart Rate Variability Analysis Using Convolutional Neural Network" for publicaton in the medicine. We are sincerely grateful for the time and effort you dedicated to reviewing our manuscript and providing valuable feedback, which has significantly improved the quality of our paper. We have taken note of the changes required and have made them accordingly within the revised manuscript. Attached, you will find a point-by-point response document addressing the reviewers' comments and concerns. We greatly appreciate your kind consideration and cooperation in approving and listing our manuscript for publication in the near future. For a detailed account of the revisions made, kindly refer to the attachment containing the revised manuscript.

Reviewer 2 Report

The manuscript is well-structured and presents a strong methodology with promising results. The use of a 1D-CNN Wavenet model for the prediction of sudden cardiac death (SCD) risk using HRV features is novel and brings a fresh perspective to this area of research. However, several points require further clarification or improvement to enhance the manuscript's quality and impact.

 The authors have done a good job of discussing ECG and its significance in SCD detection. However, the explanation of ECG might benefit from more clarity and specificity.

The authors provide a review of the current literature on HRV and SCD prediction, highlighting both the strengths and limitations of previous studies. However, some of the citations have typos such as the repeated "(2019)" on line 48.

The introduction of deep learning methods and their superiority over traditional ML techniques is well-explained. The authors might want to consider explaining why CNN specifically was chosen over other DL techniques.

The authors should better clarify the process of feature extraction and the hyperparameter tuning for the model. Explaining the logic behind choosing the range of hidden layers for grid search will be useful. Additionally, it would be helpful to clarify how you concluded that the best-performing model had seven hidden layers, a learning rate of 0.001, and a batch size of 128.

It would be beneficial if the authors could provide a graphical representation (a diagram or flowchart) of the 1D-CNN Wavenet architecture in addition to the tabular representation to enhance understanding for readers not intimately familiar with such models.

It would be useful to provide some statistics on the variance of the loss and accuracy during training and testing. Also, reaching 100% accuracy may hint at overfitting, and this point needs to be discussed.

The high performance of the model is well-argued using various performance metrics, but a discussion about the imbalance in the classes (if any) and its impact on these metrics will strengthen the manuscript.

While it's claimed that this is the first research to use CNN to extract HRV features for predicting SCD risk, a comparison with previous studies and why this method might be superior would add strength to the manuscript.

It's commendable that you used more comparison group data than other studies. However, details regarding the comparison groups, like how they were chosen and their characteristics, would be helpful for the readers.

The limitations section is appreciated, but it would be useful to add how these limitations can be mitigated in future studies. The conclusions are succinct and well-written. However, they could further benefit from highlighting the clinical implications and real-world applications of your model in the context of SCD risk prediction.

Additional Points:

Language: There are some minor grammatical errors, such as missing spaces before the citation brackets and inconsistent citation styles.

Citations: Some citation errors need to be addressed, such as duplicate years and inconsistent numbering.

Typos need to be corrected as detailed above.

Author Response

(The authors gave the same response as above.)
